# Modulation of signaling cross-talk between pJNK and pAKT generates optimal apoptotic response

**Sharmila Biswas[1]◉, Baishakhi Tikader[2]◉, Sandip Kar[2]\*, Ganesh A. Viswanathan[1]\***

**1** Department of Chemical Engineering, Indian Institute of Technology Bombay, Mumbai, India,
**2** Department of Chemistry, Indian Institute of Technology Bombay, Mumbai, India

◉ These authors contributed equally to this work.
\* sandipkar@iitb.ac.in (SK); ganeshav@iitb.ac.in (GAV)

## Abstract

Tumor necrosis factor alpha (TNFα) is a well-known modulator of apoptosis by maintaining a balance between proliferation and cell-death in normal cells. Cancer cells often evade apoptotic response following TNFα stimulation by altering signaling cross-talks. Thus, varying the extent of signaling cross-talk could enable optimal TNFα mediated apoptotic dynamics. Herein, we use an experimental data-driven mathematical modeling to quantitate the extent of synergistic signaling cross-talk between the intracellular entities phosphorylated JNK (pJNK) and phosphorylated AKT (pAKT) that orchestrate the phenotypic apoptosis level by modulating the activated Caspase3 dynamics. Our study reveals that this modulation is orchestrated by the distinct dynamic nature of the synergism at early and late phases. We show that this synergism in signal flow is governed by branches originating from either TNFα receptor and NFκB, which facilitates signaling through survival pathways. We demonstrate that the experimentally quantified apoptosis levels semi-quantitatively correlates with the model simulated Caspase3 transients. Interestingly, perturbing pJNK and pAKT transient dynamics fine-tunes this accumulated Caspase3 guided apoptotic response. Thus, our study offers useful insights for identifying potential targeted therapies for optimal apoptotic response.

## Author summary

TNFα mediated apoptosis, a form of cell-death, is the desired outcome for cancer therapeutics. This outcome is governed by complex regulation involving several signaling entities stimulated by TNFα. Relating the transient levels of these entities over early phase of the signaling response to the late-phase cell-death is a challenge. We developed a knowledge and data-driven mathematical model to unravel the dynamic synergistic correlation between the early transients governing the apoptotic response. Using flux balance and branch analysis, we demonstrated that the dynamic cross-talk between different key signaling entities regulates the apoptotic response. We performed inhibitory experiments predicted by model simulations and thereby established that cells tweak the extent of

**Data Availability Statement:** All data are in the manuscript and/or supporting information files. The GitHub repository at https://github.com/baichat90/SB-BT-Modulation-of-TNF-mediated-

crosstalk-06092022 contains the raw experimental data, the codes for parameter estimation and model analysis, and ReadMe file.

**Funding:** This study was funded by the grants CRG/2020/002672 (GAV), MTR/2020/000589 (GAV), CRG/2019/002640 (SK), and MTR/2020/000261 (SK) from Science and Engineering Research Board (SERB India; http://www.serb.gov.in), Department of Science and Technology, Government of India. The funders had no role in study design, data collection and analysis, decision to publish, or preparation of the manuscript.

synergism during signal transduction to modulate the apoptotic response in a time-dependent manner. Thus, our approach provides useful insights for identifying signaling targets to arrive at novel combinatorial cancer therapies.

## Introduction

The pleiotropic cytokine Tumor necrosis factor alpha (TNFα) mediates diverse cellular phenotypic decisions such as apoptosis, inflammation, proliferation [1–9]. In normal cells, TNFα maintains a balance between different phenotypes [10]. Since such a balance is disrupted in a diseased cell in a cancer or autoimmune milieu [11–14], its restoration using interventional therapeutic approach involving TNFα is being considered recently [15,16]. Often these strategies become ineffective to cause optimal apoptosis [17,18]. This is because TNFα activates a number of signaling pathways [19,20] culminating in varied apoptotic response. This leads to a question as how to optimally control the TNFα mediated apoptotic dynamics in a cell-type specific manner.

One of the primary events involved in cells exhibiting an apoptotic response is activation of Caspase3. Caspase3 is one of the effector caspases and is a key player involved in regulation of cell-death [11,19,21,22]. Along with modulating cell-death, TNFα strongly controls the survival responses by transducing information through entities such as NFκB [21,23–26], pAKT which is activated by PI3K [27–30]. These entities along with pJNK are well-known regulators of survival and apoptosis [31,32]. However, the extent of dynamic signaling cross-talk between these entities regulating the optimal cell-type specific apoptotic outcome remains unclear.

Mathematical modeling of molecular network [33–35] that regulates a cell's response to TNFα enables quantification of the dynamic behavior of various entities [2,36]. These network modeling frameworks provide an opportunity to investigate underlying signaling cross-talk behavior in a context-specific manner. For example, a combined computational and perturbative experimental study showed that along with other intracellular phospho-proteins, pAKT may be involved in influencing the stimulus-strength dependent kinetics of the short-term (4 h) TNFα signaling induced apoptotic response [37,38]. These studies suggest that in shorter time scales, pAKT could be a key modulator of TNFα induced apoptosis via Caspase3 activation. However, apoptosis in mammalian systems occurs over much larger timescales. Correlating these observations across timescales requires a detailed investigation of the TNFα triggered activation of Caspase3, whose early phase transients are regulated by various intracellular proteins, among which a key regulator is NFκB [39].

NFκB transients in a cell stimulated by TNFα contain sufficient information to sense the stimulus dose and accordingly fine-tune the regulation of cellular responses [40]. Recent study on NIH3T3 cells demonstrated that NFκB dynamics precisely reflects the rate of change in dose of TNFα stimulation [41]. The underlying TNFα molecular network contains several integrated pathways and multiple feedback regulations. Both quantitative and qualitative mathematical models of the network could provide a precise understanding of the dynamical properties of such a complex system. Single-cell level model of the TNFα dose-dependent NFκB signaling revealed that a cell's commitment to apoptotic or proliferative phenotype could be embedded in the early transient response of NFκB [42]. This raises a question as to how precisely the early NFκB transients refine the sustained Caspase3 activation that eventually results in a long-term apoptotic response.

Discrete-level models of TNFα network predicted the experimentally observed qualitative trend of phenotypic pro-survival and apoptotic response [43,44]. While discrete-level

modelling offers identification of signaling trends, it cannot account for time-dependent correlations which is required for unravelling the extent of cross-talk between different entities. However, predicting dynamic cross-talk and its ensuing effects on the long-term phenotypic response requires a detailed kinetic modelling. Recent kinetic modelling showed that TNFα mediated apoptosis is influenced by the TNFα level [45–47] and the overall mitochondrial mass [48]. These do not reveal the underlying dynamic signaling cross-talk between various intra-cellular intermediate entities.

Hence, the goal of this study is to decipher the dynamic signaling cross-talk between the key intracellular entities that regulates Caspase3 downstream and subsequently correlate it with the phenotypic apoptotic response. In order to achieve this objective, we took an experimental data-constrained systems biology approach. In particular, we seek to distill out this dynamic cross-talk regulating the TNFα mediated phenotypic apoptotic outcome in a context-dependent manner by considering U937 as a model cell line. The parameters involved in the kinetic model needs to be estimated, as these are not available in the literature for the considered cell-type. The methodology adopted leads to grossly identifiable sets of model parameters and thereby making our multi-input model predictive. We quantify the transient contributions by pJNK, pAKT and Caspase3, and their combinatorial effect that govern the hidden dynamic correlation using a detailed flux and branch analysis of the model of TNFα network. Our study revealed that targeted inhibitors can refine the dynamic cross-talk as well as the apoptotic outcome.

## Results

### NFκB signal inhibition enhances TNFα mediated apoptotic response

To study the dynamics of cellular fate on soluble 17.3kDa TNFα stimulation, we measured the apoptosis fraction using Annexin V assay (Methods) in the U937 cells. A 24 h exposure of TNFα (100 ng/ml) results in 25% of the population undergoing cell-death (Fig 1A, blue bar) (Details of estimation of the cell-death along with associated controls are in S1 Text). Low cell-death following exposure to only TNFα indicates that in U937 cells the cytokine might be favoring the survival signaling as well [2] and maintains a balance between the apoptotic and survival responses. As a strong mediator of survival signaling [11], NFκB is likely to be activated in response to TNFα stimulation in U937 cells. In order to ascertain it's involvement, we exposed U937 cells to Triptolide (TPL) which blocks NFκB transactivation and thereby arrested signal flow through survival pathways [49]. The relative drop in the survival response due to TPL confirms NFκB inhibition in U937 cells (S1 Fig). Pre-treatment with TPL (60 nM) and subsequent exposure to TNFα resulted in 80% cell death (Fig 1A, black bar). A comparison of apoptosis levels for this case with those corresponding to only TPL (Fig 1A, red bar) pre-treatment determines that blocking survival pathway in U937 can lead to sufficient apoptotic response. However, those cells experiencing a pre-treatment of TPL showed a greater initial (8 h) apoptotic response in the presence of TNFα (Fig 1A, red and black bars). At the same time, the levels of cell death in the case of only TPL, and those pre-treated with TPL and exposed to TNFα were similar at 24h. This indicates that the survival signaling pathway coordinates with those of apoptosis in order to regulate the cell-death in U937 cells. Overall, in the early phase (<8 hours), the apoptosis level during the combined treatment is nearly equal to the sum of that achieved for individual TPL and TNFα stimulation hints the presence of a synergistic response. On the contrary, in the later phase (>8 hours), the apoptosis levels for TPL and TPL+TNFα treatment are nearly equal. However, these responses do not reveal how signal flow through different branches of the network orchestrates this synergism.

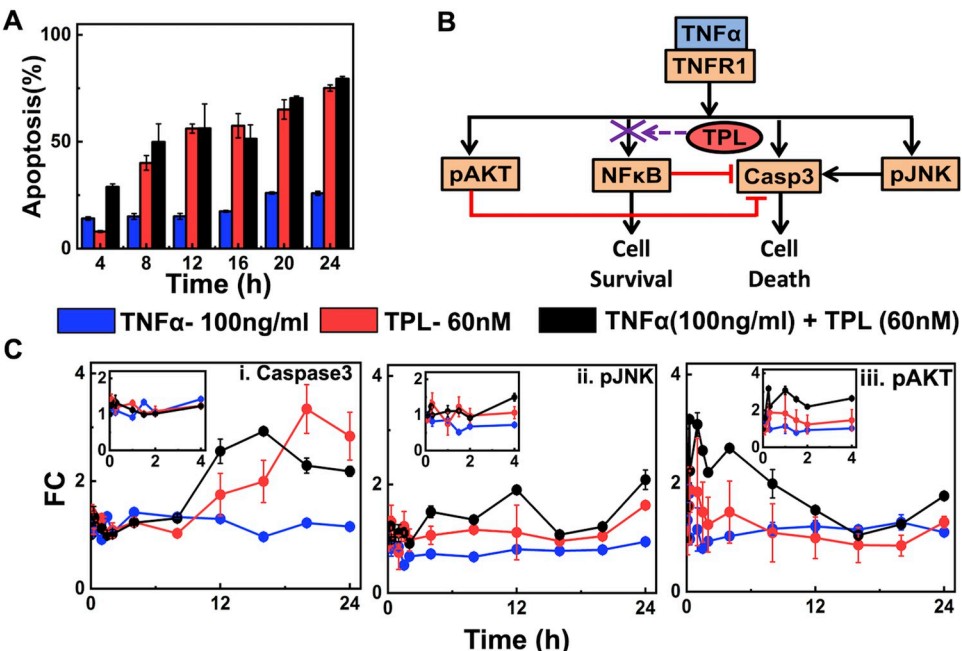

**Fig 1. Apoptosis and intracellular signaling patterns due to TNFα stimulation in the presence and absence of NFκB activity.** (A) Cell death percentage measured using Annexin V assay under three different experimental conditions at different time points. (B) A schematic of TNFα signaling mediating survival and apoptotic responses via key intracellular regulators. (C) Trajectories of relative fold change (FC), as defined by Eq (4) in Methods, of (i) Caspase3, (ii) pJNK, and (iii) pAKT for three different experimental conditions (insets show the zoomed version for the first 4 hours). Experiments were done in triplicates using U937 cells. Details of extraction of apoptosis percentage along with controls of intracellular proteins, and trajectories of untreated cells over 24h time period are presented in S1 Text.

Deciphering this synergism requires distilling out the intracellular dynamic features that govern the survival and apoptotic responses in TNFα stimulated U937 cells. The key regulatory entities involved in this orchestration are depicted in Fig 1B. It is well established that the two upstream signaling entities pAKT and pJNK regulate apoptosis [38,50,51]. pAKT has a significant role in preventing Caspase3 activation [52]. pJNK has a role in activating downstream caspases [53] and thereby regulating cellular apoptosis. This role of pJNK has been found to be context-specific [53,54]. Thus, we simultaneously measured the dynamic levels (in terms of relative fold change (FC)) of these activated signaling proteins using high-throughput experimentation [55,56] (Methods; S1 Text), which are presented in Fig 1C. In the case of only TNFα stimulation, Caspase3 dynamics being relatively unaltered for the entire duration (Fig 1Ci, blue) corroborates with the low apoptosis percentage over 24 hours (Fig 1A, blue bar). Note that even for pJNK and pAKT trajectories for the case of only TNFα stimulation, an initial increase in the fold change was observed (Fig 1Cii–1Ciii (inset), blue bar). On the other hand, under NFκB inhibitory conditions (TPL and TNFα +TPL), a rise in the later time period (>8h) of the Caspase3 FC (Fig 1Ci, red and black bars) commensurate with the enhanced cell death as found in Fig 1A (red and black bars). The subsequent continuous decrease in pAKT is reflected by a sustained increase of Caspase3 levels.

These results suggest that cells undergo enhanced apoptosis as well as a marked change in the dynamics of pAKT, pJNK and Caspase3, if NFκB mediated survival signaling is inhibited using TPL. However, the extent of dynamical crosstalk between these intracellular entities cannot be inferred from such observations. Therefore, how the dynamics of upstream proteins refine the downstream Caspase3 response, eventually regulating the apoptosis, cannot be

determined from these observations. It remains inconclusive as to which entities could be tuned to regulate the cellular decision-making events involved in cell survival or apoptosis. Moreover, the synergism in eliciting distinct apoptotic responses in the early and later phases cannot be identified using the simple network proposed in Fig 1B. Thus, to understand how the crosstalk between multiple intracellular dynamic signals quantitatively modulates cellular apoptosis, we need to develop a predictive mathematical model of the underlying TNFα network.

## Data-driven mathematical model of TNFα network

The TNFα network wiring diagram considered for the data-driven mathematical model is depicted in Fig 2A, which is an expanded version of Fig 1B. Biochemical details of the interactions in Fig 2A are presented in S2 Text and S2 Fig. The network model (Fig 2A) assumes that soluble 17.3kDa TNFα upon binding to membrane bound TNFR1 activates MAPK cascades (pERK and pJNK), NFκB and pAKT survival pathways along with the Caspase3 pathway, which mediates apoptosis [2]. (Since TNFR2 is activated only by 26kDa transmembrane TNFα, TNFR2 and its downstream signaling are excluded.) Molecular players such as ceramide are also activated upon TNFα stimulation, which subsequently influence the pAKT and pJNK levels [57]. MKK4/7, ROS and pJNK are involved in a positive feedback loop. As NFκB inhibition resulted in an augmented level of apoptosis (Fig 1), we included nodes governed by NFκB. These nodes are involved in activation or inhibition of pAKT, pJNK and Caspase3. In our model, Bcl2 and PI3K directly activate pAKT, while CAPP inhibits pAKT. TNFR1 and pJNK activate Caspase3, while pAKT and NFκB inhibits Caspase3. MKK4/7 and C1P induce the activation of pJNK, but ERK1/2 and XIAP/Gadd45β deactivate pJNK expression. We incorporated these entities into the model to extensively study the signaling cross-talk between these entities leading to apoptosis regulation [58–60]. To reduce the network complexity, we introduced direct or minimal set of interactions to account for long-range feedbacks. Interactions (Figs 2A and S2) represent either direct activation, direct inhibition or overall causal effect due to a complex set of sequential biological regulations. While some of these interactions were expressed in terms of mass-action kinetics, the remaining were modeled using either Michaelis-Menten or Hill-type kinetics to appropriately capture the underlying nonlinear behavior. (Details of each of these are given in Table I in S2 Text.)

The proposed mathematical kinetic model of the network (S1, S2 and S3 Tables) was calibrated using experimentally observed relative FC data of pJNK, pAKT and Caspase3. (Details of the calibration are given in S3 Fig) By implementing the appropriate statistical criteria (Methods; S2 Text), the kinetic parameters were estimated (S3 Table). Boxplot analysis revealed that the 3% best-fit parameter sets (out of 2000 samples) are reasonably identifiable (S4 Fig). These 3% parameter sets were used for generating model simulated trajectories, which fit well with the experimental data within the error margin, as shown in Fig 2B. (S5 Fig shows the best-fit trajectory.) Model simulations show that inhibition of NFκB by TPL cannot be restored even in the presence of a combination of TPL and TNFα (S3 Text and S6 and S7 Figs) which substantiates the similarity in the transient Caspase3 dynamics for TPL and TNFα +TPL treatments (Fig 2Bvi and 2Bix). This further shows that TPL inhibition is indeed significant in shutting down the activity of NFκB.

In order to test the predictive ability of the model, we examined the simulated transients with independent experimental observations obtained for cells pre-treated with lower concentration of TPL (10 nM) in the presence TNFα (S8 Fig). The small mean square displacements between the experimental data and simulated trajectories indicate that the model predicted pAKT, JNK and Caspase3 dynamics with reasonable accuracy (S8 Fig). Moreover, under TPL

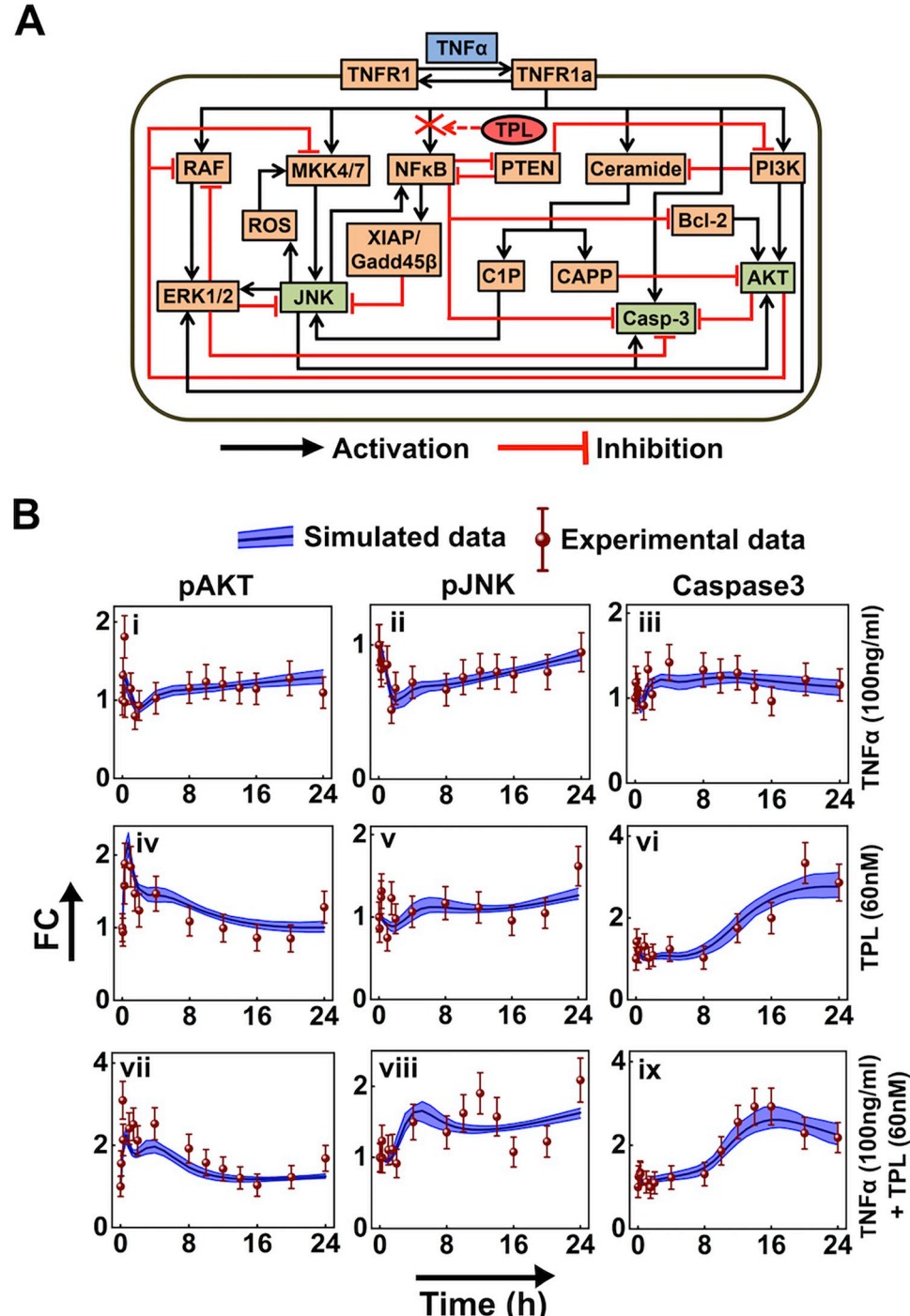

**Fig 2. Interaction network and model training with quantitative time resolved data of U937 cells.** (A) Schematic of TNFα signaling network. Black arrows and red hammers, respectively represent activation and inhibition of entities. While green boxes indicate experimentally measured marker proteins, red cross captures NFκB inhibition by TPL (red oval). Multiple pathways from NFκB inhibiting Caspase3 are lumped as a single inhibitory interaction. (Details of the entities and the interactions are in S2 Text.) (B) A comparison of the experimentally measured (red-wine circles) and model estimated signaling marker trajectory of pAKT (i, iv, vii), pJNK (ii, v, viii), and Caspase3 (iii, vi, ix) under different stimulation conditions. In each of these, black lines capture the mean of best 3% model-fitted trajectories. The blue cloud around the mean encompasses all 3% best-fitted trajectories. Note that the experimental measurements are those in Fig 1C. To constrain the model better, two additional data points (10 and 14 hours) were included during parameter estimation. The levels at these two time points were estimated using spline fitting the experimental data. Error bar around each experimental data point were calculated using standard error model.

(10 nM)+TNFα (100 ng/ml) case, analysis of the relative influence of identified cross-talk related parameters on the predicted dynamics of these entities revealed that the pAKT and pJNK dynamics are the least sensitive (S9 Fig). On the other hand, the Caspase3 dynamics is sensitive to some of the parameters to a higher extent (S9 Fig).

Overall, the mathematical model of the network reliably mimics our experimental findings. However, mere comparison of the transients of the marker proteins do not offer insights into how the overall dynamics is regulated. Such an insight can be obtained by systematically unraveling the dynamic cross-talk between various entities. Unless otherwise stated explicitly, we will use the model with the identified best parameter sets for further cross-talk analysis.

## Unraveling the dynamic cross-talk between signaling entities in the network

We next focus on unraveling the dynamic cross-talk between pAKT, pJNK and Caspase3 transients. We quantitate this by analyzing influence of specific entities in the network (Fig 2A) on these transients by capturing the corresponding biochemical reaction rate or flux. A signal flowing via an interaction from a start to an end node can be quantified by the reaction flux transduced by it. This reaction flux via an interaction is specified by the rate at which the start node influences the dynamics of the end node. Note that the instantaneous level of a certain protein is a linear combination of the contributing fluxes with appropriate sign. (Details of flux calculations along with associated expressions are provided in S4 Text.) We track the dynamic evolution of these fluxes (S10 Fig) under all three stimulation conditions (TNFα, TPL, TNFα +TPL). We present in Fig 3 the dynamic evolution of the fluxes of the interactions that contribute significantly to the transient levels of pAKT, pJNK and Caspase3.

The reaction flux analysis shows that pAKT dynamics is positively regulated by pJNK and PI3K, while CAPP is a negative regulator of pAKT under only TNFα stimulation (Fig 3Ai). In the initial phase, pAKT was majorly controlled by pJNK (Fig 3Ai), thus the drop in the pJNK contribution is reflected on the decreasing levels in the initial phase (up to 4 h) of the dynamics of pAKT (Fig 2Bi). At later time-points, both pJNK and PI3K mediated effects dominate over other fluxes, and overcome the CAPP facilitated inhibition to initiate the late phase activation of the pAKT dynamics (Figs 3Ai and 2Bi). Bcl2 protein does not influence pAKT level, as its activity is repressed by NFκB (Fig 3Ai). However, under NFκB inhibitory conditions (TPL and TNFα +TPL), Bcl2 induces a rapid increase in the reaction flux of pAKT (Fig 3Bi and 3Ci) during the initial phase (up to 2 h). This results in a sharp increase in the initial dynamics of pAKT (Fig 2Biv and 2Bvii). A comparison of the contributions to pAKT for only TNFα stimulation (Fig 3Ai) and those with TPL stimulation (Fig 3Bi and 3Ci) suggests that pJNK contributions increased relatively as a result of NFκB inhibition. However, PI3K contribution is completely lost, which can be attributed to the absence of NFκB inhibition of PTEN (Fig 3Bi and 3Ci). This re-balancing of fluxes ensures a sustained maintenance of the pAKT dynamics at the later phase (Fig 2Biv and 2Bvii). In the case of only TPL, CAPP mediated inhibition of pAKT is insignificant (Fig 3Bi) as CAPP activation via Ceramide requires TNFα stimulation. Thus, pAKT levels in the case of TPL are higher compared to that of TNFα treatment. Since pAKT dynamics is primarily governed by TPL, the TNFα+TPL treatment as well shows higher pAKT levels as compared to only TNFα case.

The reaction flux investigation further indicates that C1P and MKK4/7 are the key controllers of pJNK dynamics, while XIAP/Gadd45B (XG) negatively influences it (Fig 3Aii, 3Bii and 3Cii). Our analysis reveals that initially the inhibitory effect of XG dominates over the contribution from any other nodes under only TNFα stimulation (Fig 3Aii), which creates a steep drop in the FC of pJNK (Fig 2Bii) in the early phase. In the later phase, the C1P and MKK4/7

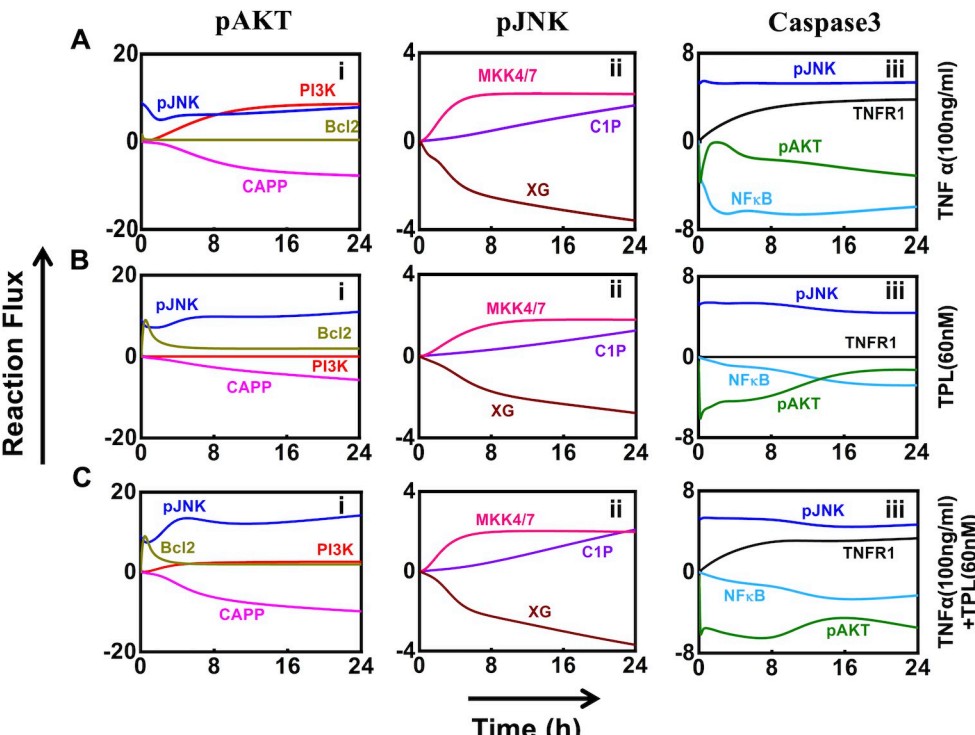

**Fig 3. The evolution of majorly contributing fluxes towards pAKT, pJNK and Caspase3.** Panels (A), (B), and (C), respectively show the evolution of fluxes for the three markers under the stimulation conditions TNFα, TPL, and TNFα + TPL. In each of the panels (i), (ii), and (iii), respectively show the evolution of fluxes contributing to dynamics of pAKT, pJNK and Caspase3 dynamics for the three different stimulation conditions. The individual reaction fluxes were estimated using the rate equation described in S4 Table. Evolution of all contributing fluxes has been presented in S10 Fig.

primarily control the levels of pJNK (Fig 3Aii). However, in the presence of TPL, the inhibition by XG appears to be less effective (Fig 3Bii and 3Cii), which helps to stimulate the dynamics of pJNK at early time points (Fig 2Bv and 2Bviii).

Flux analysis further unfolds that NFκB and pAKT are the major negative regulators of Caspase3 dynamics, while pJNK and TNFR1 are its positive modulators (Fig 3Aiii, 3Biii and 3Ciii). When stimulated with only TNFα (Fig 3Aiii), the balance between the predominant inhibition by NFκB and mutual activation mediated by TNFR1 along with pJNK aid in maintaining sustained levels of Caspase3 (Fig 2Biii). Under TPL treatment, the initial dynamics of Caspase3 seems to be influenced both negatively by pAKT and positively by pJNK (Fig 3Biii) resulting in no activation of Caspase3 at the early phase (~8h) (Fig 2Bvi). As the later phase depicts a reduced pAKT inhibitory contribution towards Caspase3 flux (Fig 3Biii), there is an augmentation in the level of Caspase3 dynamics (Fig 2Bvi). Under the influence of TPL in the absence (Fig 3Biii) and presence (Fig 3Ciii) of TNFα, the contribution from NFκB reduced significantly, while pAKT plays a predominant role in maintaining the dynamics of Caspase3 (Fig 2Bix). The inhibition mediated by pAKT was relatively less effective and thereby, controls the threshold extent of Caspase3 protein resulting in a sudden increase in its level (S10 Fig). Note that the positive contribution via pJNK helps in maintaining the sustained levels of Caspase3 under all three stimulation conditions (Fig 3Aiii, 3Biii and 3Ciii). However, for TNFα +TPL condition, the negative contribution of pAKT on Caspase3 dynamics starts increasing after 10h (Fig 3Ciii), leading to a gradual decrease in the Caspase3 levels (Fig 2Bix) during the later phase (16–24 h).

In summary, there is a signaling cross-talk between the pAKT and pJNK dynamics which in coordination with NFκB regulates the Caspase3 dynamics. However, our flux analysis does not clearly identify how the signals originating from the TNFR1 and due to TPL pre-stimulation are channeled through various branches in order to orchestrate the pAKT and pJNK transients mediated synergistic signaling response to Caspase3.

## Branch analysis reveals the synergistic dynamic cross-talk signaling regulating Caspase3 transients

We next perform a systematic branch analysis to distil out the extent of synergism exhibited in the dynamics of the signaling entities (pAKT and pJNK) based on the three stimulation conditions. First, we identify the branches originating from TNFR1 or NFκB and ending in pAKT or pJNK (Tables 1 and S5). We analyze the relative capacity to process the signal flow through these branches for an $i^{th}$ stimulation condition, where $i$ stands for TNFα, TPL, or TNFα+TPL. The relative capacity ($\mathbb{R}$) of a branch, say, NFκB→pAKT ($B_{na1}$, Table 1), to process signal during $i^{th}$ stimulation is given by

$$\mathbb{R}_{NF\kappa B \to pAKT}|_i = (J_{NF\kappa B \to pTEN} \times C_{pTEN})_i \times (J_{pTEN \to PI3K} \times C_{PI3K})_i \times (J_{PI3K \to pAKT} \times C_{pAKT})_i \quad (1)$$

where, $J_{l \to k}$ captures the absolute flux flowing along the interaction from $l$ to $k$. $C_k$, the relative flux processing capacity of node $k$, is given by

$$C_k = \frac{\sum_{\forall n,m} J_{n \to m}}{\sum_{\forall p,n} J_{p \to n}} \quad (2)$$

where $n$, $m$ and $p$ is an index for the entities of the network (Fig 2B). We define extent of synergism ($\mathbb{S}$) facilitated by $B_{na1}$ as

$$\mathbb{S}_{NF\kappa B \to pAKT} = \frac{\mathbb{R}_{NF\kappa B \to pAKT}|_{TNF\alpha+TPL}}{\mathbb{R}_{NF\kappa B \to pAKT}|_{TNF\alpha} + \mathbb{R}_{NF\kappa B \to pAKT}|_{TPL}} - 1. \quad (3)$$

Note that, for a given branch, $\mathbb{S} > 0$ and $\mathbb{S} < 0$, respectively indicate positive and negative synergism. We estimated the extent of synergism for all branches in Tables 1 and S5.

**Table 1. Major branches originating from NFκB or TNFR1 and ending in the signaling entities pAKT or pJNK.** Comprehensive list of all branches are presented in S5 Table.

| ID | Branch |
|----|--------|
| $B_{na1}$ | NFκB ⊣ PTEN ⊣ PI3K⟶AKT |
| $B_{na2}$ | NFκB⟶XG ⊣ JNK⟶AKT |
| $B_{na3}$ | NFκB ⊣ PTEN ⊣ PI3K⟶ERK1/2 ⊣ JNK⟶AKT |
| $B_{ta1}$ | TNFR1a⟶PI3K⟶AKT |
| $B_{ta2}$ | TNFR1a⟶RAF⟶ERK1/2 ⊣ JNK⟶AKT |
| $B_{ta3}$ | TNFR1a⟶Ceramide⟶C1P⟶JNK⟶AKT |
| $B_{nj1}$ | NFκB⟶XG ⊣ JNK |
| $B_{nj2}$ | NFκB ⊣ Bcl−2⟶AKT ⊣ MKK4/7⟶JNK |
| $B_{nj3}$ | NFκB ⊣ PTEN ⊣ PI3K⟶Ceramide⟶C1P⟶JNK |
| $B_{nj4}$ | NFκB ⊣ Bcl−2⟶AKT⟶RAF⟶ERK1/2 ⊣ JNK |
| $B_{tj1}$ | TNFR1a⟶RAF⟶ERK1/2 ⊣ JNK |
| $B_{tj2}$ | TNFR1a⟶Ceramide⟶C1P⟶JNK |
| $B_{tj3}$ | TNFR1a⟶Ceramide⟶CAPP⟶AKT ⊣ RAF⟶ERK1/2 ⊣ JNK |

We present the dynamic variation of the extent of synergism facilitated by major branches (Table 1) in Fig 4. Branch $B_{na1}$ originating from NFκB offers positive synergism in the early phase and switches to negative synergism beyond 8 hours (Fig 4A, red). In particular, this positive synergism corresponds to the early rise of pAKT transient response for the TNFα+TPL case (Fig 2Bvii), which could be attributed to the effect of TPL via PI3K. On the other hand, the negative synergism in the later phase correlates with the drop in the pAKT response induced by TPL. While branch $B_{na2}$ is responsible for maintaining the pAKT levels in the later phase, $B_{na3}$ enables the early phase contributions of pJNK to pAKT transients (Fig 4A, blue and green). The undulations in the pAKT transients (Fig 2B) are due to the dynamic variation in the extent of positive synergism via branches $B_{ta2}$ and $B_{ta3}$, both originating from TNFR1 (Fig 4B, gray and purple). These undulations are perhaps caused by the positive feedback loop involving RAF, ERK and JNK embedded in $B_{ta2}$ (Table 1). The early phase switch in the synergism enabled by the branch $B_{ta1}$ highlights the TNFα mediated signaling to pAKT via PI3K (Fig 4B, turquoise).

The steady marginal negative synergism via branch $B_{nj1}$ is responsible for maintaining the JNK levels at later time points (Fig 4C, dark brown). On the other hand, the time-dependent undulated positive synergism observed in branches $B_{nj2}$ and $B_{nj3}$ (Fig 4C, light brown and dark blue) is manifested in the pJNK transients (Fig 2Bv and 2Bviii). The presence of RAF-ERK-JNK positive feedback in branch $B_{nj4}$ (Table 1) could have resulted in the multiple rise and drop of synergism (Fig 4C, dark green). The nature of synergistic dynamics observed in $B_{nj4}$ indicates a weak representation of this branch in the overall pJNK transient. This is due

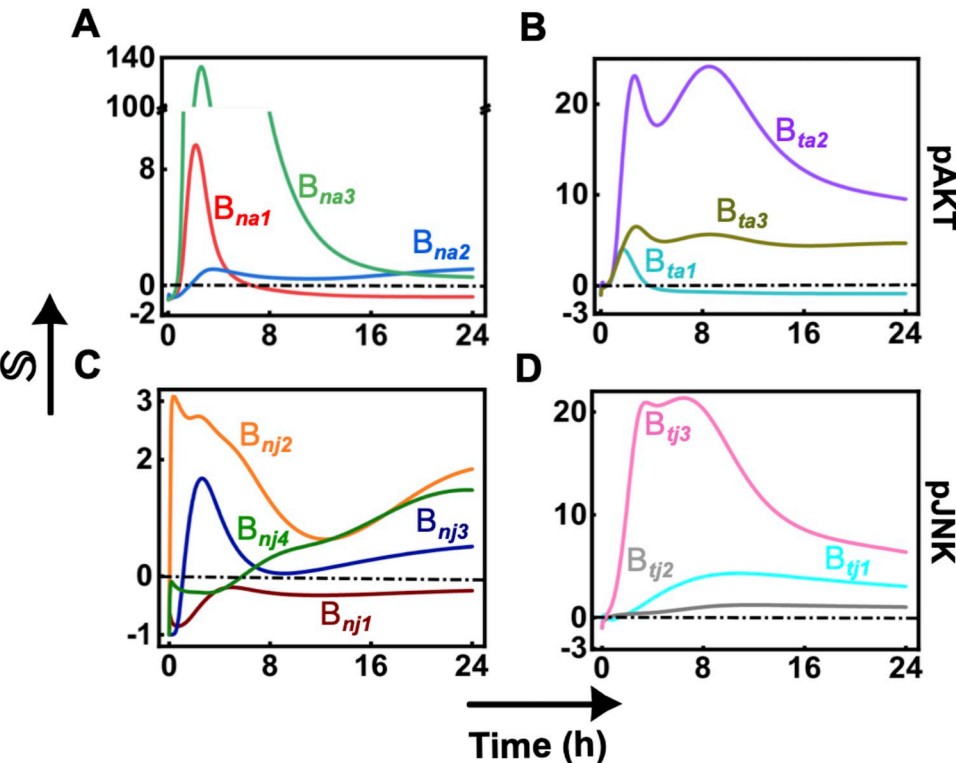

**Fig 4.** Dynamic evolution of the extent of synergism ($\mathbb{S}$, defined in Eq 3) facilitated by major branches as depicted in Table 1. Synergism ($\mathbb{S} > 0$ is positive and $<0$ is negative) quantified from major branches originating from NFκB to (A) pAKT and (B) pJNK, and that from TNFR1 to (B) pAKT and (D) pJNK. The evolution for other branches (S5 Table) are shown in S11 Fig.

to the poor estimation of the kinetic parameters (such as $K_{mae}$, $K_{eir}$, and $K_{air}$) involved in the RAF-ERK-JNK feedback loop, which could explain the qualitative difference between the model fitting and experimental observations for pJNK under TNFα+TPL condition. Signal via branch $B_{tj1}$ and $B_{tj2}$, both originating from TNFR1 maintain the initial basal pJNK transient levels in a synergistic manner (Fig 4D, turquoise and pink). Moreover, the signal flow via branch $B_{tj3}$ contributes to keeping the transient pJNK levels high at the later phase (Fig 4D, dark blue). Contributions due to the remaining branches are elucidated in S5 Text and S11 Fig.

Overall the branch analysis elucidates how the transient pAKT and pJNK are governed by signal flow through different major branches in the TNFα signaling network (Fig 2A). This further confirms that a simple coarse-grained network in Fig 1A cannot capture the intricate dynamic signaling cross-talk and the synergism orchestrating the pAKT and pJNK dynamics. However, since branch analysis compares the relative signal flow due to multiple stimulation conditions, it cannot be employed to identify how the Caspase3 transient dynamics is precisely controlled by the pAKT and pJNK. In order to decipher this control, we next perform a time-dependent correlation analysis under the three stimulation conditions.

## Dynamic correlation between pJNK and pAKT levels modulates overall Caspase3 transient response

The cooperative influence of pJNK and pAKT over Caspase3 transients need not necessarily be instantaneous and could be over an extended period of time. pAKT and pJNK accumulated over an extended period may coordinate in a time-dependent manner to control the overall Caspase3 levels along with the apoptotic response. We perform a systematic time-dependent correlation analysis of the accumulated levels of pAKT and pJNK to decipher the underlying regulation under different stimulation conditions. The accumulation of pAKT and pJNK are quantified by estimating the Area Under the Curve ($AUC$) of their transient responses (Methods) over different phases of the dynamics. In particular, we consider accumulation from the start (0 h) of the stimulation up to 8h, 12h and 24h for the correlation analysis. Note that the transients generated using the model for 5 best fit parameter sets were employed for this purpose.

The average relative contributions of accumulated pAKT ($A_{pAKT}$) and pJNK ($A_{pJNK}$) to overall Caspase3 levels are defined respectively by Eqs (7) and (8) in Methods. The relative contributions for the three stimulated conditions are presented in Fig 5A. When cells were stimulated with just TNFα, pJNK primarily contributes to the Caspase3 accumulated over various time durations (Fig 5Ai, shaded bars). Under TPL condition, pAKT dominates the Caspase3 accumulation for the first 8h (Fig 5Aii, unfilled bars). On the other hand, pJNK, which promotes Caspase3 activation, contributed negatively to Caspase3 accumulation (Fig 5Aii, shaded bars). Accumulation up to 12h shifted the pJNK contribution from negative to positive, while pAKT contribution decreased (Fig 5Aii). Subsequently, the influence of these two entities on overall Caspase3 level up to 24h becomes poised (Fig 5Aii). In the case of TNFα +TPL, the positive contribution from pJNK towards Caspase3 accumulation over 8, 12 and 24h increases along with a decrease in the impact of pAKT on it (Fig 5Aiii). This indicates that pJNK plays a significant role in regulating Caspase3 expression throughout the entire time course under all three stimulation conditions except during the initial phase for only TPL treatment.

Since Caspase3 along with other effector caspases such as Caspase7 are known to initiate apoptotic response [22], we determine how apoptosis is reflected in the overall Caspase3 accumulation ($AUC_{casp3}$) under these three stimulation conditions (Fig 5B). First, we analyze the

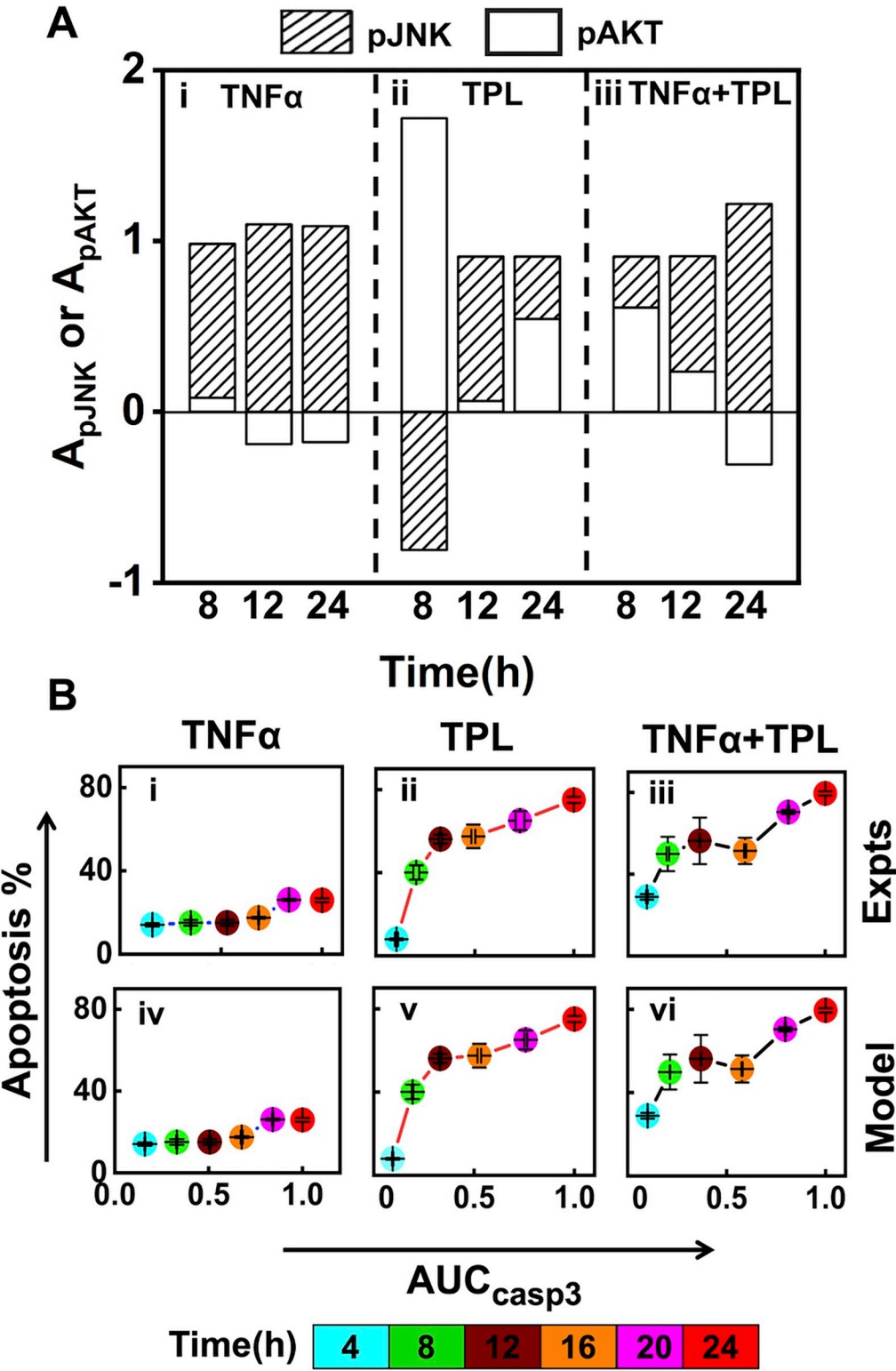

**Fig 5. pJNK and pAKT transients synergistically control the overall Caspase3 levels and modulate the apoptotic response.** (A) Relative contribution of model generated pJNK and pAKT transients corresponding to 5 best fit parameter sets ($A_{pJNK}$ and $A_{pAKT}$) towards the corresponding overall Caspase3 accumulation for 8, 12 and 24 h durations for (i) TNFα, (ii) TPL, (iii) TNFα +TPL stimulation. (B) Depicts the correlation of experimentally measured accumulated Caspase3 levels ($AUC_{casp3}$) (n = 3) with corresponding apoptosis percentage for the stimulation conditions

(i) TNFα, (ii) TPL, (iii) TNFα +TPL. (iv-vi) represent correlation of model predicted accumulated Caspase3 levels ($AUC_{casp3}$) (n = 5) with the experimentally measured apoptosis percentage (n = 3) for the three stimulation conditions. The time points are appropriately color-coded in (B).

experimentally measured apoptosis levels (Fig 1A) with the corresponding accumulated Caspase3 (Fig 1Ciii). For TNFα stimulation condition, the apoptosis percentage did not rise concomitantly with the increasing $AUC_{casp3}$ (Fig 5Bi). This could probably be due to the activation of the survival pathway (NFκB), which interferes with the apoptosis process (Fig 2A). When the NFκB mediated survival signals are interrupted using TPL, there was an enhancement of Caspase3 accumulation over a 24h time period with an increase in the apoptosis level (Fig 5Bii). Moreover, the proportion of cells undergoing apoptosis increased monotonically with Caspase3 accumulation until 12 hours, following which no substantial rise in apoptosis (50% to 80%) occurred with an increase in $AUC_{casp3}$ (Fig 5Bii). Similar behavior was observed for the TNFα +TPL case as well (Fig 5Biii). A comparison of the early phase apoptosis response for the TPL (Fig 5Bii) and TNFα +TPL (Fig 5Biii) cases shows marginally higher percentage apoptotic response as that in the former. This suggests that Caspase3 accumulation up to a particular threshold at the early time point (4-12h) is a pre-requisite to commit cells for apoptosis, which is only moderately affected by the Caspase3 surge at later time points (24h) for U937 cells. Next, we show the experimentally observed apoptosis levels with the model-predicted overall Caspase3 accumulation (Fig 5Biv–5Bvi). A comparison of Fig 5Bi–5Biii and 5Biv–5Bvi demonstrates that the model simulated $AUC_{casp3}$ predicts the apoptosis levels as good as that by those from experimental measurements. This suggests that even though the accumulation information was *not* considered in the model training process (Fig 2B), the transients generated using the estimated model parameters adequately captures the apoptosis response via $AUC_{casp3}$. In summary, both pJNK and pAKT, and their coordinated cross-talk with the NFκB regulation finetunes the apoptotic response by modulating the overall Caspase3 levels.

## Inhibiting pAKT and pJNK signaling alters Caspase3 transient response

Dynamic correlation analysis (previous section) suggests that fine-tuning the apoptotic response could be achieved by altering the overall Caspase3 dynamics by perturbing the pAKT and pJNK transients. We introduce these perturbations by exposing U937 cells to either Wortmannin (Wort) or SP600125 (SP6). Wort inhibits PI3K mediated pAKT activation (Fig 6A) [61,62], pJNK activation through MKK4/7 [63–65] and XG activation through NFκB [66]. On the other hand, SP6 affects both pJNK and pAKT activation via different pathways (Fig 6B) [67]. SP6 further activates Caspase3 [68] by down regulating Bcl2 activation leading to reduction in pAKT levels and inhibits MKK4/7 through the AP-1 and Fas driven pathway [67,69–73]. We incorporate these perturbations using phenomenological terms in the model, which will henceforth be referred to as *inhibitory model*. (Details of inhibitory model are described in S6 Text.)

We first consider the case of Wort inhibition in our model simulations, which demonstrate that after a transient initial peak, pAKT levels decrease with time under all three stimulation conditions (Fig 6Ci). This could be due to pJNK playing a significant role in activating pAKT (Fig 2A). On the contrary, for all three stimulation conditions, after an initial dip, pJNK dynamics showed a steady increasing trend with an undulation (Fig 6Cii). This could be due to the RAF-ERK-JNK feedback loop identified using the branch analysis (Fig 4C and 4D and Table 1). For the case of only TNFα, in the presence of Wort, model predicts a marginal increase in Caspase3 level (Fig 6Ciii, blue cloud). However, when stimulated only with TPL or TNFα+TPL, simulations showed a gradual time-dependent increase in the Caspase3 levels

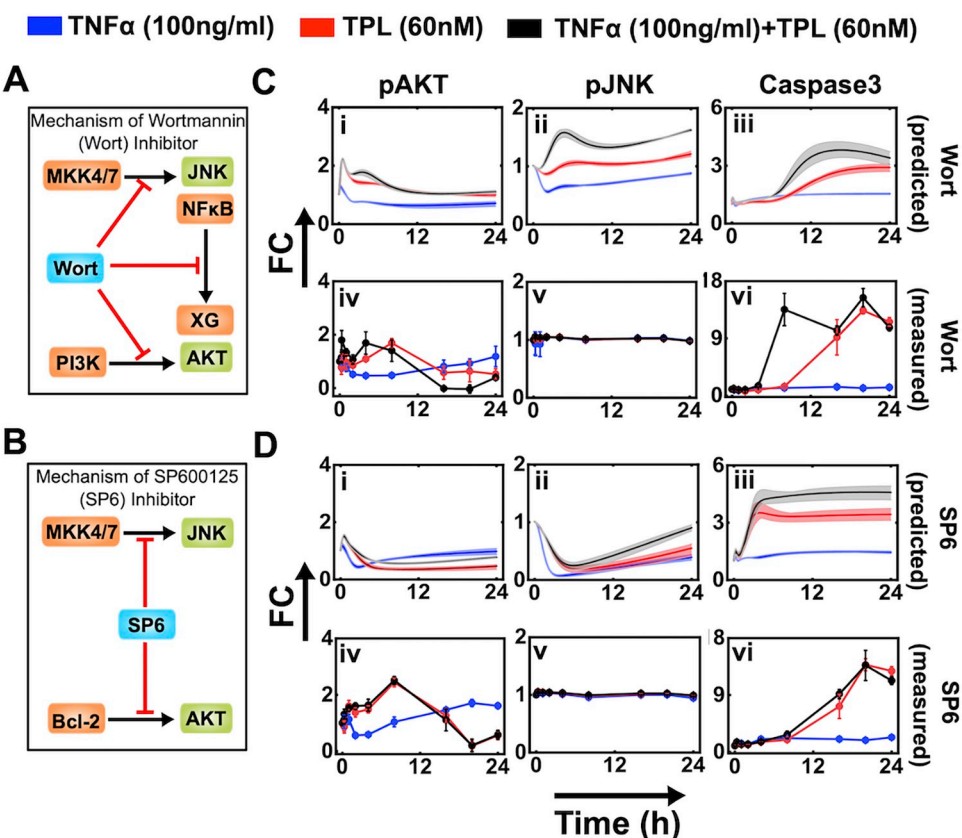

**Fig 6. Model prediction and experimental validation of the marker proteins under the treatment of Wortmannin (Wort) and SP600125 (SP6) inhibitor.** (A) and (B), respectively schematically represents the inhibitory action of Wort and SP6. (C) Effect of Wort inhibition on (i) pAKT, (ii) pJNK, and (iii) Caspase3 as predicted by model simulations. The corresponding experimental validations are in (iv), (v), and (vi), respectively. (D) Effect of SP6 inhibition on (i) pAKT, (ii) pJNK, and (iii) Caspase3 as predicted by model simulations. The corresponding experimental validations are in (iv), (v), and (vi), respectively. The rate constant values used for the inhibitory interactions are given in the caption of S12 and S13 Figs.

from the initial phase (Fig 6Ciii, red and black clouds). The flux analysis (S12 Fig) demonstrates that the effect of Wort reduces the inhibitory contributions of pAKT and NFκB fluxes over Caspase3 dynamics. For all three stimulation conditions, the levels reached in the initial phase subsequently settles into the corresponding steady level by 24 hours (Fig 6Ciii). To validate these model predictions, we measured the transient levels of pAKT, pJNK and Caspase3 in Wort treated U937 cells under all three stimulation conditions (Fig 6Civ–6Cvi). (A comparison of the model predictions and the corresponding measurements with those obtained for the case of no Wort inhibition is presented in S6 Text.) The experimentally measured pAKT and Caspase3 dynamics corroborated qualitatively with the theoretical model predictions (Fig 6Ci, 6Ciii, 6Civ and 6Cvi). However, the sustained very low fold-change detected experimentally for pJNK could not be predicted by the model simulations (Fig 6Cii and 6Cv), which we attribute to poor estimation of certain kinetic parameters related to a major branch involving RAF-ERK-JNK feedback loop.

We next investigate the effect of SP6 inhibition. The model predicted trend for pAKT, pJNK and Caspase3 were by and large similar to those obtained for the case of Wort inhibition for respective stimulation conditions (Fig 6Di–6Diii). (Inhibitory model predictions along with the experimental measurements under three different stimulation conditions are

contrasted with those obtained for the case sans SP6 inhibition in S12 Fig) The experimentally measured dynamics of these proteins in SP6 treated U937 cells qualitatively substantiate the model predictions for all three stimulation conditions (Fig 6Div–6Dvi). Specifically, for the TPL and TNFα+TPL stimulation conditions, the rise in the Caspase3 levels were delayed as compared to model simulations (Fig 6Diii and 6Dvi).

Flux analysis of the inhibitory model simulations (S6 Text) revealed that in the presence of Wort, weak inhibition of JNK activation by MKK4/7 was required to capture the qualitative experimental trend of Caspase3 and pAKT transients. Activation of JNK is primarily caused by MKK4/7 and C1P under normal conditions (Fig 3Aii, 3Bii and 3Cii). Under Wort inhibitory conditions, the fluxes to JNK from these interactions almost remained unaltered (S12 Fig, ii, v, viii) leading to poor model prediction of pJNK dynamics. Experiments show that transient Caspase3 rise correlates with drop in the pAKT transient under TPL and TNFα+TPL conditions. However, for these two conditions, in the model predictions, the pAKT transients are maintained at a higher steady-level, which is above that for the case of only TNFα treatment. This is the reason why the model underestimates the relative rise in the Caspase3 transients for TPL and TNFα+TPL treatment under Wort inhibitory conditions. Flux due to pJNK maintains a higher pAKT level at later time points (S12 Fig, i, iv, vii). On the other hand, in the case of SP6 mediated inhibition, the flux analysis reveals that MKK4/7 mediated pJNK activation is completely shut down causing the early phase dip in pJNK and thereby in pAKT transients as well (S13 Fig, ii, v, viii). As a consequence, the model predicts the rise in the Caspase3 dynamics occurring much earlier than that observed in experiments (Fig 6Diii). Moreover, for TPL and TNFα+TPL cases, except TNFR1 mediated Caspase3 activation, all other fluxes contribute to Caspase3 transients in a similar manner (Fig 6Diii). This substantiates the experimentally observed fact (Fig 6Div) that the Caspase3 transients are similar for TPL and TNFα +TPL treatments.

Overall, our model could qualitatively predict the modulation in Caspase3 dynamic response under all the three experimental conditions (TNFα, TPL and TNFα +TPL) in the presence of either Wort or SP6 inhibitor as substantiated by the experimental data. Thus, model is able to predict the cell-fate decision in U937 cells for which the $AUC_{casp3}$ can act as a suitable intracellular signature. This leads to the hypothesis that fine-tuning apoptosis can be achieved by controlling $AUC_{casp3}$.

## Model adequately predicts apoptotic response in U937 cells under all stimulation conditions

In order to test the above hypothesis, we contrast the model predicted apoptotic response with those observed experimentally when U937 cells are treated with Wort or SP6 inhibitors under all three stimulation conditions (TNFα, TPL, TNFα +TPL). For predicting apoptosis level using the model simulated $AUC_{casp3}$ at a certain time, we assume the correlation between $AUC_{casp3}$ and apoptosis in Fig 5Biv–5Bvi as a calibration for the three stimulation conditions. (Note that the inhibitory action via Wort or SP6 are on signaling entities upstream of Caspase3. Therefore, the relationship obtained between the $AUC_{casp3}$ and apoptosis under noninhibitory conditions (Fig 5Biv–5Bvi) would continue to hold under inhibitory conditions as well.) In order to find the apoptotic response for the entire range of $AUC_{casp3}$ in Fig 5Biv–5Bvi, we quantify the correlation between model generated $AUC_{casp3}$ vs experimentally observed apoptosis using a polynomial curve fit. (Details of the fit and identifiability of the associated parameters are provided in S7 Text and S6 Table.) Note that the curve fitting was performed using the $\langle AUC_{casp3} \rangle$, which is an average over that estimated using five best-fit model simulated trajectories. Next, using the Wort or SP6 inhibitory model simulations, for a certain time

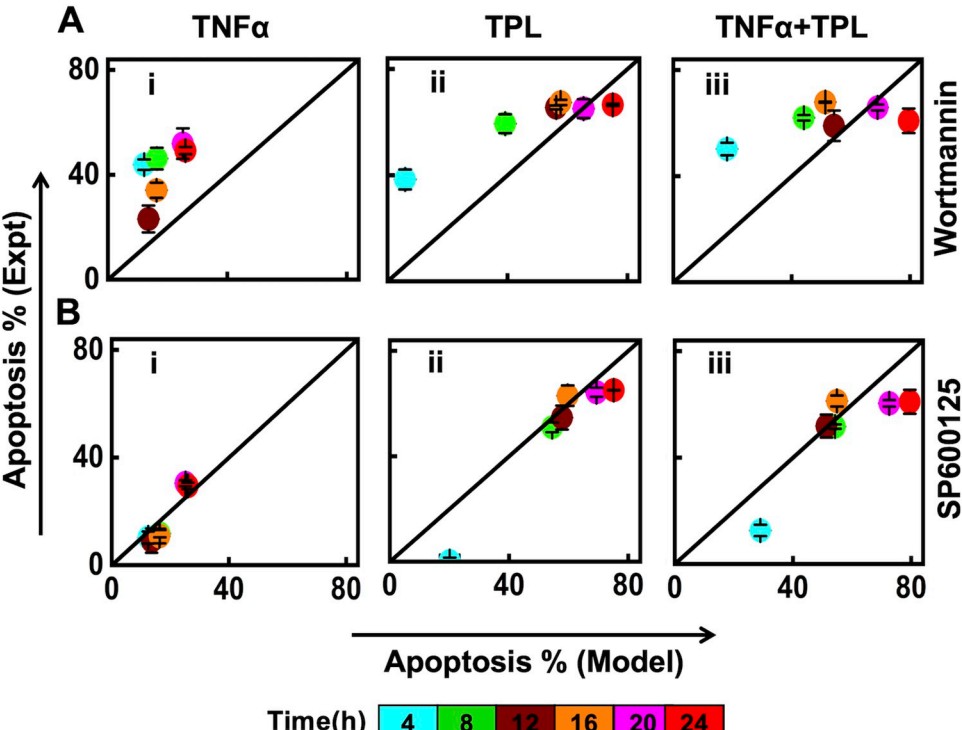

**Fig 7. Comparison of experimental and model predicted apoptosis percentage under (A) Wort and (B) SP6 inhibitory conditions.** (i), (ii), and (iii), respectively correspond to the predictions for TNFα, TPL, and TNFα +TPL stimulation conditions. While 5 best fit parameters were considered for model predictions, those of experiments are based on triplicates. The time points are appropriately color-coded in (B).

duration under a specific stimulation condition, we estimated the $\langle AUC_{casp3}\rangle$, which was incorporated in the polynomial fit to arrive at the model predicted apoptosis percentage (Fig 7). The corresponding apoptosis percentage in U937 cells was measured experimentally (Methods).

For the Wort inhibition case under only TNFα stimulation condition, the model underpredicted the experimentally observed apoptosis percentage (Fig 7Ai). This is due to poor prediction of the pJNK level by the Wort inhibitory model (Fig 6Aii, blue). This deviation is reflected in the lower levels of Caspase3 transient (Fig 6Aiii, blue) causing decreased $\langle AUC_{casp3}\rangle$. For the TPL and TNFα +TPL cases (Fig 7Aii–7Aiii), the Wort inhibitory model predicts the experimental apoptosis percentage at all time durations other than 4h. Note that after 16h the experimental apoptosis levels become insensitive to the marginal increment in $\langle AUC_{casp3}\rangle$ under both these conditions (Fig 7Aiii).

The SP6 inhibitory model predicted apoptosis levels match the experimental apoptotic measurements in U937 cells for all the time points during only TNFα stimulation (Fig 7Bi). For the case of TPL and TNFα +TPL stimulations, for intermediate time points (8, 12, 16h), the experimental observations corroborate model predictions (Fig 7Bii and 7Biii). However, for these two conditions, the model over-predicts the apoptosis levels at earlier and later duration. The over-prediction at the early time point (4 h) could be due to model predicting significantly higher levels of initial Caspase3 transients as compared to the experimental observations (Fig 6Biii and 6Biv). Experimentally observed apoptosis response being insensitive to accumulated Caspase3 levels (Fig 7Bii and 7Biii) could explain the overprediction by the model at later time points (Fig 6Diii). Juxtaposition of model predicted $\langle AUC_{casp3}\rangle$ and

apoptosis leading to a correlation can help contrasting the experimentally observed and model simulation predicted apoptotic cell-fate response. Such a comparison can be achieved by inhibiting other entities in the network as well.

## Discussion and conclusion

Dynamic crosstalk among intracellular signaling entities plays a crucial role in regulating the cell-fate as a response to the pleiotropic cytokine TNFα stimulation. In this study, we present an experimental data-constrained mathematical model consisting of 17 entities and 25 interactions that can predict the apoptotic response under different physiological conditions related to TNFα signaling network. The experimental data represents the dynamical response due to activation of signaling pathways downstream of TNFR1 stimulated by the soluble 17.3 kDa TNFα. (Note that signaling due to TNFR2, activated only by 26 kDa transmembrane TNFα, and its downstream signaling are ignored.) Using systems biology-based approaches, we unravel the synergistic interdependence of pAKT and pJNK transient response and their dynamic crosstalk modulating apoptosis via Caspase3 dynamics in U937 cells in a semi-quantitative manner.

While TNFα signaling maintains a delicate balance between the survival and apoptotic responses [10], tilting the balance towards apoptosis by inhibiting signaling downstream NFκB has been demonstrated [74]. There are several interconnected signaling branches that could influence this cell-death decision [75]. Our experimental findings unfold that arresting survival signaling via NFκB pathway using Triptolide (TPL) aided in quantifying the transient influence of pJNK and pAKT activity on the apoptotic response in U937 cells (Figs 1 and 5A). The delayed Caspase3 response observed in our experiments and predicted by the model is in line with those reported in other cell lines [76]. By monitoring NFκB and Caspase signaling for just 1–2 hours, Lee *et al.* revealed that a 1-min TNFα pulse can be more efficient at killing cells than a 1-hour pulse [46]. Timescale of apoptotic response and the upstream signaling influencing the same is cell-type specific. Our model analysis revealed that the nature of influence of the transients of the signaling entities on cell-death is time-dependent.

For U937 cells, we identified that the influence in the early phase ($< 8$ hours) is distinct to that in the late phase (8 to 24 hours) (Fig 8). This was deciphered by tracking the nature of synergism in signal flow via different branches using the branch analysis (Fig 4). For example, the signal transduction requires NFκB→pAKT branch to dynamically switch from a positive to negative synergism across the two phases (Fig 8). The low Caspase3 levels at the early phase and the subsequent rise in the late phase (Fig 2) can be attributed to the early phase negative synergism in the NFκB→pJNK branch being absent in the late phase (Fig 8). Moreover, the positive synergism elicited by the branches originating from TNFR1 involving the RAF-ERK1/2-JNK positive feedback loop influences the dynamics of pAKT in both early and late phases (Fig 8). Combined effect of the dynamically varying synergism in these branches governs the Caspase3 transients via pAKT and pJNK, and therefore the phenotypic apoptotic response (Fig 5B).

Traditional approach of discrete-level modeling identified that pJNK and pAKT can influence the TNFα mediated apoptotic response but does not lead to precise extent of dynamic cross-talk [43,44]. The reaction flux analysis along with the branch analysis revealed the causal mechanism regulating apoptotic response involving the dynamic cross-talks among the key intracellular entities pJNK and pAKT (Figs 3 and 4). We identified that pJNK transient response and PI3K play crucial roles in controlling pAKT dynamics. While PI3K primarily influences the late-phase activation of pAKT, pJNK affects it over the entire 24 h duration (Fig 3). Such a different early and late-phase responses of the marker proteins have also been observed during p53 signaling induced by DNA damage response [77–80].

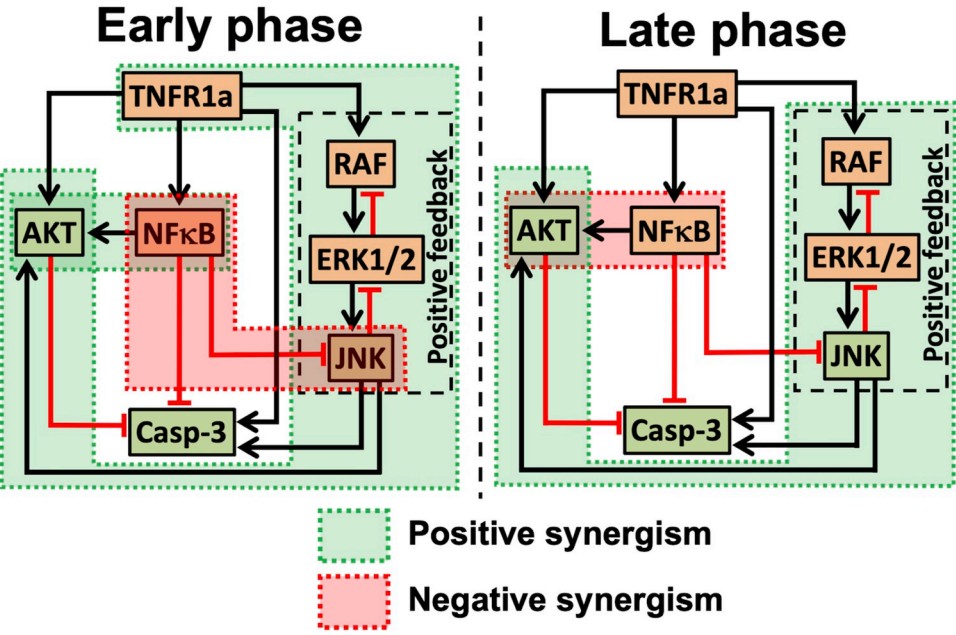

**Fig 8. Comparison of time-dependent synergism during the early and late phase signaling cross-talk regulating Caspase3 activation.**

Since cell-death is a long-term phenotypic response, identifying the quantitative relationship between the intracellular marker levels and the apoptosis can offer strategies for its modulation. Contrary to that observed experimentally, TNFα network model developed by Koh and Lee predicted that Caspase3 activation predominates at short timescales post-stimulation [81]. This could be attributed to not considering pJNK, which is a crucial Caspase3 regulator, as shown by our predictions (Fig 5A). While arresting the pro-survival pathways is expected to alter the cell-death decisions [82,83], our study demonstrates that a semi-quantitative correlation establishes the connection between the Caspase3 dynamics and the apoptotic response (Fig 5B). The premise of this correlation is to relate the Caspase3 levels over a range of time duration represented by accumulated levels to the phenotypic response. Therefore, the correlation can serve as a guide for modulating the TNFα mediated cell-death behavior.

Correlation between the model simulated accumulated Caspase3 levels and experimentally detected apoptosis response enabled prediction of apoptotic phenotype in the presence of Wortmannin and SP600125 inhibitors (Fig 6). Even though the phenotypic predictions are valid only in the range of the accumulated Caspase3 levels, the systematic semi-quantitative approach employed can be easily extended to other perturbative conditions. While this approach can help gain insights into various therapeutic responses, there is a scope for further refining the model for improving its predictive abilities. For example, the model did not adequately predict the pJNK transient behavior (Fig 2). This can perhaps be reconciled by introducing additional pJNK regulating nodes or interactions. Cell-to-cell variability is inherently present even in our experimental measurements (S1 Text). However, our model, due to its deterministic nature, is not appropriate to mimic such variabilities. Recent studies have indeed shown that superimposing stochasticity on a deterministic model could allow accounting for cell-to-cell variability [45,84]. Thus, the model proposed here can be extended to capture the ensemble-level behavior. Overall, we demonstrate that predicting the synergistic cross-talk signaling guided qualitative apoptotic phenotypic response is possible even without capturing the detailed pathway of apoptosis in an explicit manner.

## Materials and methods

### M1. Cell culture and reagents

U937 cells were procured from the Cell Repository at National Centre for Cell Science (NCCS), Pune, India. It was cultured in RPMI-1640, supplemented with 10% fetal bovine serum (FBS), 2 mmol/l L-glutamine, and 1% antibiotic–antimycotic solution, all procured from HiMedia (Mumbai, India) with a cell-seeding density of $5\times10^5$ cells/ml. Cells were then collected, pelleted by centrifugation at 1,000 rpm for 5 min at room temperature (RT) and maintained at 37˚C in a humidified 5% $CO_2$ incubator. 17.3kDa TNFα (Peprotech) was reconstituted in double-distilled water to a concentration of 100μg/ml. Triptolide (TPL) (Sigma) was dissolved in DMSO to a concentration of 1 mg/ml, the stock was stored at −20˚C until use. Antibodies against different proteins ((phospho-pAKT(pS473)- Alexa 488-tagged antibody, phospho-pJNK(pT183/pY185)- PE tagged, and anti-active Caspase3- V450 tagged antibody)) were procured from BD Biosciences.

### M2. Apoptosis detection using Annexin V/PI staining

U937 cells were plated in 24-well plates. For a certain experimental condition, cells in different wells were treated with the corresponding stimulation for 4, 8, 12, 16, 20 or 24 h. For those conditions involving TPL, cells were pre-treated with it for 1 h prior to TNFα addition. After the completion of the stimulation time, cells were harvested, washed once with PBS, resuspended in 1X Annexin binding buffer and stained with FITC-labeled Annexin V and PI (BD Pharmingen, San Diego, CA, US). Cells were then analyzed on BD FACS Aria (BD Biosciences, San Jose, CA, US) within 30 min of dye addition. For every experimental condition and time point, three replicates were used.

### M3. Intracellular staining with Flowcytometry using Fluorescent Cell Barcoding (FCB)

U937 cells were seeded in 12 well plates for measuring signaling levels at 12 timepoints over a span of 24 h. For both TPL and TNFα +TPL stimulation conditions, cells were pre-treated with TPL for 1hr. After the stimulation, cells were fixed with 4% PFA for 10 min at RT, washed once with PBS and then transferred to a 96-well plate for further processing. Cells were then permeabilized with 500μl of 80% chilled methanol on ice for 1 h, and washed in PBS to remove leftover methanol. Now 50μl of 4 different concentrations (10μg/ml, 2μg/ml, 0.4μg/ml and 0.08μg/ml) of barcoding dye (Alexa-647) was added to the samples (450μl in PBS) making up the total volume to 500μl and kept on ice for 40min. After the incubation, cells were washed with 0.5%BSA in PBS to remove unbound dye. Finally, a combo tube was prepared that consisted of 4 different concentrations of the dye and washed thrice with 0.5%BSA in PBS to remove excess unbound dye. Cells resuspended in 0.5% BSA in PBS, were stained with antibodies against pAKT, pJNK and Caspase3 with dark incubation at RT for 30min. After removing the unbound antibody, cells resuspended in 300 μl of 0.5% BSA were analyzed for detecting fluorescence in various channels on a BD FACS Aria flow cytometer. A schematic of this entire high-throughput protocol is in Fig 9.

### M4. Data quantification

The obtained file was then deconvoluted using FlowJo (version 10). Each fluorochrome used corresponds to a particular protein. Mean Fluorescence Intensity (MFI) of the distribution was used to calculate the relative fold change of the protein level upon stimulation. Relative

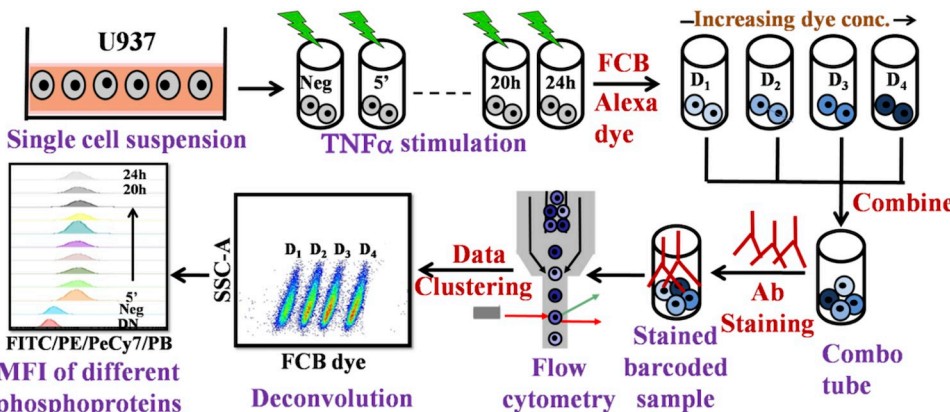

**Fig 9. Schematic of high throughput signaling experimentation.** Cells are harvested and incubated with inhibitor or stimulus for the desired time. After incubation, cells are fixed using paraformaldehyde and subsequently permeabilized with methanol. Four samples, each containing sufficient number of cells, are then incubated with amine-reactive fluorescent barcoding (FCB) [55,56] dye with different concentrations. These four samples are combined in a combo tube. After covalent binding, cells in the combo tube are washed several times to remove unbound dye, following which they are exposed to monoclonal antibody against the intracellular proteins. Fluorescence emitted by cells in the combo tube is then acquired on a flow cytometer. After acquisition, the samples are analyzed by gating and identifying individual samples displaying discrete fluorescent intensities (D1, D2, . . . D4) in the FCB channel. Mean fluorescent intensities of the fluorescence distribution corresponding to individual proteins are captured in different channels based on the fluorochrome antibody for further analysis.

fold change (FC) at a certain time point is given by

$$FC = \frac{MFI_t - MFI_{DN}}{MFI_{Neg} - MFI_{DN}} \tag{4}$$

where, $MFI_t$, $MFI_{Neg}$, $MFI_{DN}$, respectively represent MFI at a certain time $t$, negative control (no stimulation (0 h)), and double negative (only cells).

## M5. Signaling pathway perturbation with inhibitors

For each of the stimulation conditions, cells were treated with inhibitor Wortmannin (1μM) or SP600125 (10 μM) for 1 h prior to stimulation. After treatment, cells were stained as per the method specified earlier. Further the fluorescence emitted by the stained samples were acquired using BD FACS Aria following which FC at the measured time points were estimated using Eq (4).

## M6. Model kinetic parameter estimation

The model kinetic parameter estimation was performed using the PottersWheel software (version 4.1.1) [85] by employing "Trustregion" method with underlying ODEs integrated using CVODE method. The model system has been optimized ~2000 times by taking the same initial starting values of the parameters (parameters are globally fitted by considering a logarithmic parameter space) to fit the experimental data adequately. We considered $\chi^2$ value for a certain parameter set $P$ as the fitting criteria, where $\chi^2$ is defined as

$$\chi^2(P) = \sum_{i=1}^{N} \left( \frac{y_{pAKT,i}^e - y_{pAKT}(t_i; P)}{\sigma^2} \right)^2 + \sum_{i=1}^{N} \left( \frac{y_{pJNK,i}^e - y_{pJNK}(t_i; P)}{\sigma^2} \right)^2$$
$$+ \sum_{i=1}^{N} \left( \frac{y_{Casp3,i}^e - y_{Casp3}(t_i; P)}{\sigma^2} \right)^2 \tag{5}$$

where, $y^e_{pAKT,i}$, $y^e_{pJNK,i}$, and $y^e_{Casp3,i}$ represent the mean of the triplicate experimental FC data for the pAKT, pJNK, Caspase3, respectively at the $i^{th}$ time point. $y_{pAKT}(t_i; P)$, $y_{pJNK}(t_i; P)$, and $y_{Casp3}(t_i; P)$ are the levels of the respective proteins at the $i^{th}$ time point obtained from model simulations for a chosen parameter set $P$. $\sigma$ is the standard deviation of the experimental data set. The detailed description of the model development is in S2 Text.

## M7. Reaction Flux analysis

For a certain node such as pAKT, the rate law corresponding to every incoming and outgoing interactions were evaluated at every time point. For a certain interaction, a locus of these over 24 h is its reaction flux trajectory. We repeated this procedure to find the reaction flux trajectories for the interactions in which pAKT, pJNK and Caspase3 are involved. Details of these are in S4 Text.

## M8. Branch identification

TNFα signaling network (Fig 2A) was converted into an unsigned, interaction digraph. Adjacency matrix for the digraph was generated. Subroutine 'allpaths.m' in Matlab was used to identify all paths (or branches) between a pair of entities. Subsequently, for every branch, interactions in it were assigned their respective sign (Tables 1 and S5 as specified in the original network.

## M9. Correlation analysis and Linear regression

For a certain stimulation condition, area under the curve ($AUC$) was calculated for pAKT, pJNK and Caspase3 transients up to 8, 12, and 24 h time durations. For model-based $AUC$, transients generated using 5 best parameter sets were employed. Linear regression (in R [86]) was carried out using the $AUC$ from all 5 transients to find the constants $a$ and $b$ in overall pJNK and pAKT contributions in accumulated Caspase3 captured in

$$AUC_{casp3} = a \times AUC_{pJNK} + b \times AUC_{pAKT} \tag{6}$$

Relative contributions were estimated by normalizing all terms in Eq (3) with $\langle AUC_{casp3} \rangle$, the average of $AUC_{casp3}$ across replicates. Thus, the average relative contributions of pAKT ($A_{pAKT}$) and pJNK ($A_{pJNK}$), respectively are defined as

$$A_{pAKT} = \langle \frac{b \times AUC_{pAKT}}{\langle AUC_{casp3} \rangle} \rangle \tag{7}$$

and

$$A_{pJNK} = \langle \frac{a \times AUC_{pJNK}}{\langle AUC_{casp3} \rangle} \rangle \tag{8}$$

where, $\langle . \rangle$ represents average across replicates. Note that $A_{pJNK} + A_{pAKT} = 1$.

## Supporting information

**S1 Text. Apoptosis and intracellular marker protein level detection.**
(PDF)

**S2 Text. Detailed model of the TNFα signaling network capturing the cross-talk between different entities.**
(PDF)

**S3 Text. Prediction and sensitivity analysis of transient dynamics of different entities from the proposed model.**
(PDF)

**S4 Text. Reaction flux analysis of the three marker proteins.**
(PDF)

**S5 Text. Branch analysis quantifying the synergistic dynamic cross-talk signaling.**
(PDF)

**S6 Text. Model analysis under inhibitory conditions—Wortmannin (Wort) and SP600125 (SP6).**
(PDF)

**S7 Text. Semi-quantitative relationship between $\langle AUC_{casp3} \rangle$ and Apoptosis levels.**
(PDF)

**S1 Fig. Effect of TPL on cell survival.** (A) Dose-response of TPL on cell-survival. (B) Effect of TPL pre-treatment duration on apoptosis. Note that negative Apoptosis % in (B) for a few cases are due to negative control (only cells) being more than that when treated.
(TIFF)

**S2 Fig. Detailed TNFα signaling network.** Activation and inhibitory actions are represented by solid (black) and dashed (red) lines, respectively. Green and orange boxes, respectively represent the inactive and active forms of an entity. The model contains 34 species and 81 parameters including 3 scaling constants.
(TIFF)

**S3 Fig. Generalized methodology to identify best suitable model based on goodness of fit.** The proposed models were mapped with experimental data and robust model was selected by performing parameter optimization and associated identifiability analysis.
(TIFF)

**S4 Fig. Boxplot showing deviation of the estimated kinetic parameters around its median.**
(TIFF)

**S5 Fig. Model trajectories for best fitted parameter set with experimental measurements.** The best-fitted trajectories (red) of ~2000 fits with the experimental FC (circles with appropriate error bars) for pAKT, pJNK and Caspase3 under the three stimulation conditions. The errors are estimated by using a standard error model.
(TIFF)

**S6 Fig. Model predicted time course simulation of fold activation (FC) of transient varying different concentrations of TNFα.** Simulated trajectory of FC by varying TNFα in (A) the absence of TPL, (B) presence of low dose TPL (10nM) and (C) presence of high dose TPL (60nM).
(TIFF)

**S7 Fig. NFκB dynamics under different stimulation conditions.** Trajectories of NFκB protein obtained by simulating the model with best-fit parameter set (S4 Table).
(TIFF)

**S8 Fig. Comparison of model predicted transients with experimental measurements for TPL-10nM in the presence of TNFα (100ng/ml).** The green lines represent the simulated

trajectories and the black dots with corresponding error bars (n = 3) indicate the experimental measurements.
(TIFF)

**S9 Fig. Relative influence of the cross-talk related parameters on the predicted dynamics for the case of TPL-10nM in the presence of TNFα (100ng/ml).** $\Phi_k$ and $\Phi_{WT}$, respectively captures the area under the transient for the case of deviation in a certain cross-talk parameter and that for the case of the best fit parameter. Note that a deviation of 20% (as specified in the third column of Table I in S3 Text) from the best fit values in Table I in S3 Text was considered for this analysis.
(TIFF)

**S10 Fig. Temporal evolution of fluxes contributing to the dynamics of pAKT, pJNK and Caspase3 under the three stimulation conditions.** Rows correspond to different stimulation conditions. Expression for these fluxes ($A_i$, $J_i$ and $C_i$, for all $i$) are provided in S4 Table.
(TIFF)

**S11 Fig. Time-dependent synergism** due to all branches from (A) NFκB to pAKT, (B) TNFR1 to pAKT, (C) NFκB to pJNK, and (D) TNFR1 to pJNK as listed in S5 Table.
(TIFF)

**S12 Fig. Evolution of fluxes from important entities controlling the dynamics of pAKT, pJNK and Caspase3 under different experimental conditions in the presence and absence of Wort inhibitor.** Flux analysis of different nodes on controlling of pAKT, pJNK and Caspase3 under different experimental conditions in the presence of Wort inhibitor. The dotted line represents when simulation has been done at 0 nM (no inhibitor) of Wort and the solid line depicts the trajectories for 1000 nM of Wort. The contribution from each specific node in the time profile of marker protein has been dissected separately by considering various terms in the corresponding model equation in S4 Table. Wherever necessary, the inhibitory action related modifications to the relevant rate terms were considered. Inhibitory parameters used are $K_{iak}$ = 0.001 nM$^{-1}$, $K_{kim}$ = 0.02 nM$^{-1}$, and $K_{kxg}$ = 0.00005 nM$^{-1}$.
(TIFF)

**S13 Fig. Evolution of fluxes from important entities controlling the dynamics of pAKT, pJNK and Caspase3 under different experimental conditions in the presence and absence of SP6 inhibitor.** Flux analysis of different nodes on controlling of pAKT, pJNK and Caspase3 under different experimental conditions in the presence of SP6 inhibitor. The dotted line represents when simulation has been done at 0 nM (no inhibitor) SP6 and the solid line depicts the trajectories for 10000 nM of SP6. The contribution from each specific node in the time profile of marker protein has been dissected separately by considering various terms in the corresponding model equation in S4 Table. Wherever necessary, the inhibitory action related modifications to the relevant rate terms were considered. Inhibitory parameters used are $K_{kib}$ = 0.0002 nM$^{-1}$ and $K_{kis}$ = 0.005 nM$^{-1}$.
(TIFF)

**S1 Table. Ordinary differential equations of the TNFα signaling network and associated algebraic relations.**
(PDF)

**S2 Table. Description of the entities in the TNFα signaling network model and its state.**
(PDF)

**S3 Table. Definition of the kinetic parameters involved in the model.**
(PDF)

**S4 Table. Reaction fluxes affecting the pJNK, pAKT and Caspase3 dynamics.**
(PDF)

**S5 Table. Branches originating from NFκB or TNFR1 and ending in the signaling entities pAKT or pJNK.**
(PDF)

**S6 Table. Coefficients of the fourth order polynomial (Eq S7.1 in S7 Text) under different stimulation conditions.**
(PDF)

## Acknowledgments

We gratefully acknowledge an access to the IIT Bombay FACS Central Facility and BSBE FACS Facility. SB and BT thank IIT Bombay for their fellowship.

## Author Contributions

**Conceptualization:** Sandip Kar, Ganesh A. Viswanathan.

**Data curation:** Sharmila Biswas, Baishakhi Tikader.

**Formal analysis:** Sharmila Biswas, Baishakhi Tikader, Sandip Kar, Ganesh A. Viswanathan.

**Funding acquisition:** Sandip Kar, Ganesh A. Viswanathan.

**Investigation:** Sharmila Biswas, Baishakhi Tikader, Sandip Kar, Ganesh A. Viswanathan.

**Methodology:** Sharmila Biswas, Baishakhi Tikader, Sandip Kar, Ganesh A. Viswanathan.

**Project administration:** Sandip Kar, Ganesh A. Viswanathan.

**Resources:** Sharmila Biswas, Baishakhi Tikader, Sandip Kar, Ganesh A. Viswanathan.

**Software:** Sharmila Biswas, Baishakhi Tikader.

**Supervision:** Sandip Kar, Ganesh A. Viswanathan.

**Validation:** Sharmila Biswas, Baishakhi Tikader, Sandip Kar, Ganesh A. Viswanathan.

**Visualization:** Sharmila Biswas, Baishakhi Tikader, Sandip Kar, Ganesh A. Viswanathan.

**Writing – original draft:** Sharmila Biswas, Baishakhi Tikader.

**Writing – review & editing:** Sandip Kar, Ganesh A. Viswanathan.

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
