## [Decision Letter · Decision Letter 0]

8 Jul 2022

Dear Dr. Viswanathan,

Thank you very much for submitting your manuscript "Modulation of signaling cross-talk between pJNK and pAKT generates optimal apoptotic response" for consideration at PLOS Computational Biology.

As with all papers reviewed by the journal, your manuscript was reviewed by members of the editorial board and by several independent reviewers. In light of the reviews (below this email), we would like to invite the resubmission of a significantly-revised version that takes into account the reviewers' comments.

We cannot make any decision about publication until we have seen the revised manuscript and your response to the reviewers' comments. Your revised manuscript is also likely to be sent to reviewers for further evaluation.

Sincerely,

Jing Chen

Guest Editor

PLOS Computational Biology

Douglas Lauffenburger

Deputy Editor

PLOS Computational Biology

Reviewer's Responses to Questions

**Comments to the Authors:**

Reviewer #1: In this study, Biswas et al. studied signaling cross-talks between pJNK and pAKT pathways on cell apoptosis decision of a U937 cell line using data-driven mathematical modeling and quantitative measurements. The authors have performed extensive studies with various analysis methods to provide mechanistic information on how the two pathways may synergetically affect Caspase3 level and therefore affects cellular apoptosis. The results are in general solid and have the potential to expand our understanding of this important cell fate decision process.

However I feel that the presentation of this work needs substantial revision to convey more effectively on the message the authors want the readers to receive. Here are some main comments.

1) Despite all the extensive analyses, at the end I can’t find a distilled mechanistic summary that can provide an intuitive and transparent picture to general readers on how the transient dynamic profiles of the two pathways fine-tune the Caspase3 activation. On page 6, paragraph: For later time, the apoptosis levels are similar for TPL treatment and TPL + TNFα treatment; but for earlier time (4h, and to a less extent 8h), the combined treatment leads to an apoptosis level more or less the sum of individual TPL and TNFα treatment. I think this observation is the phenomenon that the authors want to address. Am I correct?

As indicated in Fig. 1B, TNFα triggers both death and survival paths (through four parallel pathways), so the final decision is a tug-of-war between the two sub-programs. TPL treatment tips the balance towards the death fate by weakening/blocking the survival program---but it should be only partially since inhibition on Caspas3 is through either pAKT or NFkB. Is this simplified network sufficient to explain the results? How (or can) the temporal dynamics of (i.e., propagation along) each branch explain the crosstalk and resultant apoptosis level?

They performed TPL treatment since it simplifies the analyses on the pAKT and pJNK pathways by blocking the NFkB pathway. It may help readers by explicitly spelling this purpose.

Do the temporal profiles shown in Fig. 2B provide clue on the “synergy” (? The terms means that the response from a combined treatment is larger than the sum of responses of individual treatments)? What molecular species serve as the converging nodes? While they used the detailed wiring diagram in Fig 2A for model studies, is it really necessary to use such a detailed model? Even so, can they also provide a more coarse-grained network structure just for understanding, so one can have a more transparent picture on how the network structure leads to specific form of the Caspase3 dynamics they observed? Feedback and feed-forward loops, bifurcation analyses, etc., which may help on the understanding? Is the network in Fig. 1B sufficient to summarize the basic structure of the network in Fig. 2B?

2) For Fig.1Cii-iii, the case of TNFα stimulation only, the authors state that there is an initial increase in the fold change in pJNK and pAKT levels (represented in blue bars). First of all, the trajectories are too packed/cramped in which the increases in fold change are really hard to see (the color bars are tangled together especially in the initial 0-8h). Also, with TNFα and TPL treatment (black bars) in Fig.1Ciii, why are the levels of pAKT in TPL only and TPL + TNFα experiments that high compare to TNFα only treatment (since the level of pAKT is not affected by TPL and the level of NFκB)?

3) In Fig. 5 where the U937 cells were treated with either Wort or SP6, the authors state that the experimentally measured pAKT levels and Caspase3 levels match well with their modeling results. However, for the TPL only and TNFα + TPL cases, the modeling results do not seem to correlate well with that in experimental data. Also, there is a discrepancy between the modeling results and experimental results on pJNK levels, as also stated by the authors themselves, that the “Wort inhibitory model underestimated the pJNK level (Fig. 5Dii)”. There might be a typo when the authors refer to this discrepancy, as they said the Wort inhibitory model underestimates the pJNK levels, and use Fig 5Aii, blue as its reference. Do they refer to Fig. 5Dii, blue, since Fig 5A is a schematic representing the Wort inhibitory mechanism. The discrepancy is also present in the SP6 inhibitory model (Fig. 5Dii and Fig. 5Dv). It would be better if the authors can explain more on the discrepancy between the modeling results and the experimental results (e.g. why did over- or underestimation happen)

For the inhibitory models along with inhibitory experiments shown in Fig. 5, although Wort and SP6 inhibit both pAKT and pJNK activation through different pathways, their functions are basically the same as the goal is to inhibit pAKT and pJNK activations. Is it necessary to perform two separate modeling and experimental studies considering their blocking targets are the same?

4) The authors need to provide sufficient details so the work is reproducible. For example, what is the cell seeding confluency? I can’t find a (supplemental) table listing all the reagents (esp. antibodies). Will the computer codes be publically available?

5) Can the authors elaborate on the statement “inhibiting pJNK and pAKT transient dynamics optimizes this accumulated Caspase3 guided apoptotic response”? Being optimal relative to what? How can one tell some optimal results were obtained? Moreover, when performing the three experiments (TNFα, TPL only and TPL + TNFα+ TPL), what states/situations were the cells at? Under stress, starving, or healthy seeding environments? Would these factors interfere with the final cellular fate decisions?

There are some more specific comments on the writing. The authors should use simple language, and correct typos.

1) Page 5, paragraph right above “Results” section: The kinetic parameters required for the detailed model considered are usually unavailable

2) Page 6, last paragraph: “no significant difference in Caspase3 dynamics”---difference between what? Cells before and after treatment?

3) Page 8: “In our model Bcl2 and PI3K directly activate the pAKT in contrast CAPP inhibit pAKT”? --Can’t understand this sentence.

4) Page 8: “These mediating crosstalk between the marker proteins leading to apoptosis regulation were incorporated”---Consider rewording, such as “ We incorporated these mediating….”

5) Fig 2B: Need to make it clear whether the simulated results were obtained from fitting the experimental data, or predicted with parameter values estimated independently from the data. For the former, one should label it clearly as “Model fitting results”

6) Page 11: One or two sentences explaining reaction flux will be helpful to general readers to get intuitive understanding of the quantity

7) Page 11, second paragraph: as its activity is repressed by NFκB

8) Fig 5 C &D: It might be clearer to indicate predicted and measured results on the figure.

9) Page 21: There are several parallel pathways that could influence this cell-death (decision?)

Reviewer #2: This manuscript utilized an experiment-informed mathematical model to investigate the crosstalk of intracellular signals and Caspase3 activation and for cell apoptosis, as well as how they are mediated by NFκB. The authors demonstrate the predictive power of this model in an independent condition (Figure S9) and when the cells were treated with inhibitors (Figure 5). The authors further used this model to predict cell apoptosis under different inhibitors conditions. Overall, the model was well examined and offered insights into the interplay of intracellular signals. I have several suggestions:

1. It is not well justified that the correlation between apoptosis and AUCcasp3 (Figure 4) can persist in the same way for different inhibitors conditions and can thus be applied to study other inhibitors, as presented in Figure 6. Can the authors offer some explanations for that?

2. The data comparison between the model and experiments in Figure 4B is almost identical. I am worried about a potential overfitting of the model. Can more analyses be done to justify the transferability of the model in this case?

3. In the section “Data-driven mathematical model of TNFα network”, the authors mention, “Since TNFR2 is activated only by 26kDa transmembrane TNFα, TNFR2 and its downstream signaling are not considered in the model”. More discussion, perhaps in the Discussion and Conclusion section, will be helpful in explaining the exclusion of TNFR2 for regulating cell fate.

4. References 2 and 57 refer to the same article.

5. Some figures in SI text are mislabeled, for example, Figure S4 in S2.4.

6. Some labels are missing in Figure S5.

Reviewer #3: In this manuscript, the authors made both experimental and simulation studies on the optimal apoptotic response, i.e., the modulation of signaling cross-talk between pJNK and pAKT. In the experiments, the authors observed that the inhibitor of NF-kB, TPL, can promotes apoptosis remarkably in the presences or absence of TNF-α. Then, they developed a data-driven model to reveal the mechanism in cellular response to TNF-α. They can reproduce the time courses of the key signaling markers by mathematical modeling. They attempted to unravel the mechanism underlying the crosstalk between the JNK and AKT pathways from different perspectives. Moreover, they found that inhibiting pJNK and pAKT transient dynamics optimizes this accumulated Caspase3-dependent apoptotic response. Their results are very relevant, and also valuable for further study. However, the presentation of the manuscript is not very clear, and should be improved largely, and also some more discussions should be added. Especially, some comments as the following should be considered.

1. NF-kB was mainly considered as a prosurvival factor in this work. The time courses of NF-kB or its targets should be shown experimentally in different cases to reflect the activities of NF-kB. The comparison of NF-kB activities is helpful of validating the inhibitory effects of TPL on NF-kB. Moreover, similar comparison is also required in the simulation results.

2. According to their model (Fig. 2A), it was assumed that JNK can activate NF-kB. How does this regulation influence apoptosis induction in the presence NF-kB inhibition by TPL.

3. The dose of TNF-α was fixed at 100ng/ml. How does the doses of TNF-α modulate the crosstalk between signaling pathways and apoptosis induction?

4. In Figs. 5C and 5D, there exists obvious bias between the simulation and experimental results for pJNK, the authors should give an explanation to this point.

5. The duration of two treatments in the combinatorial therapies may results in distinct cellular outcomes (PMID: 26965628). How does the duration between TPL pre-treatment and TNF-α exposure influence cell fates?

6. In this work, the marker proteins showed different regulation modes in the early and late phase of the response. In the DNA damage response, p53 also exhibits different regulation effects on apoptosis in the early and late phase of the response (PMID: 29048401; PMID: 21576488; PMID: 19617533). The shared principles in cellular response to different stimuli should be discussed.

7. The formats of the items in the Reference should be unified.

8. The English writing of the manuscript should be improved. For example, page 10, the bottom paragraph, “is” in “how the overall dynamics is regulated” should be corrected; page 15, the sentence “which is only moderately affected by ……” should be removed to page 14 to follow closely with the last sentence; page 21, the second line from the bottom, “connect” should be connection. Of note, the list is not exhausted, the authors should check the writing of the manuscript entirely.

**Have the authors made all data and (if applicable) computational code underlying the findings in their manuscript fully available?**

Reviewer #1: **No: **I could not find the statement in the manuscript

Reviewer #2: **No: **The authors have provided the details of their methods, but the code is not available publicly.

Reviewer #3: **No: **Please provide all the related data and code

PLOS authors have the option to publish the peer review history of their article (what does this mean?). If published, this will include your full peer review and any attached files.

Reviewer #1: **Yes: **Jianhua Xing

Reviewer #2: No

Reviewer #3: No
---

## [Decision Letter · Decision Letter 1]

3 Oct 2022

Dear Dr. Viswanathan,

We are pleased to inform you that your manuscript 'Modulation of signaling cross-talk between pJNK and pAKT generates optimal apoptotic response' has been provisionally accepted for publication in PLOS Computational Biology.

Best regards,

Jing Chen

Guest Editor

PLOS Computational Biology

Douglas Lauffenburger

Section Editor

PLOS Computational Biology

Congratulation! Please note Reviewer 2's comments on the resolution of your main figures and make sure you upload high-quality figures in the production stage.

Reviewer's Responses to Questions

**Comments to the Authors:**

Reviewer #1: The authors have addressed my concerns.

Reviewer #2: The authors have addressed my previously raised questions. I appreciate their efforts. They need to check the resolution of their main text figures.

Reviewer #3: Congratulations!

**Have the authors made all data and (if applicable) computational code underlying the findings in their manuscript fully available?**

Reviewer #1: Yes

Reviewer #2: Yes

Reviewer #3: None

PLOS authors have the option to publish the peer review history of their article (what does this mean?). If published, this will include your full peer review and any attached files.

Reviewer #1: **Yes: **Jianhua Xing

Reviewer #2: No

Reviewer #3: No

---

## [Editor Report · Acceptance letter]

9 Oct 2022

PCOMPBIOL-D-22-00784R1 

Modulation of signaling cross-talk between pJNK and pAKT generates optimal apoptotic response

Dear Dr Viswanathan,

I am pleased to inform you that your manuscript has been formally accepted for publication in PLOS Computational Biology. Your manuscript is now with our production department and you will be notified of the publication date in due course.

With kind regards,

Zsofia Freund
